# How Does Contrastive Pre-training Connect Disparate Domains?

## Abstract

Pre-training on massive unlabeled datasets greatly improves accuracy under distribution shifts. As a first step toward understanding this, we study a popular pre-training method, contrastive learning, in the unsupervised domain adaptation (UDA) setting where we only have labeled data from a source domain and unlabeled data from a target domain. We begin by showing on 4 benchmark vision datasets that out-of-the-box contrastive pre-training (even without large-scale unlabeled data) is competitive with other UDA methods. Intuitions from classical UDA methods such as domain adversarial training focus on bringing the domains together in feature space to improve generalization from source to target. Surprisingly, we find that contrastive pre-training learns features that are very far apart between the source and target domains. How then does contrastive learning improve robustness to distribution shift? We develop a conceptual model for contrastive learning under domain shifts, where data augmentations form connections between classes and domains that can be far apart. We propose a new measure of connectivity —the relative connection strengths between same and different classes across domains—that governs the success of contrastive pre-training for domain adaptation in a simple example and strongly correlates with our results on benchmark vision datasets.

## 1 Introduction

In applications such as image recognition for self-driving cars (Yu et al., 2020; Sun et al., 2020) or medical image diagnosis (AlBadawy et al., 2018; Dai & Gool, 2018), machine learning models often fail on examples drawn from a different distribution than training. Pre-training on large-scale unlabeled data (Chen et al., 2020; He et al., 2020; Radford et al., 2021) can substantially boost accuracy under distribution shift (Radford et al., 2021; Hendrycks et al., 2019; Fisch et al., 2019; Yogatama et al., 2019; Devlin et al., 2019). For example, CLIP (Radford et al., 2021), which is trained on internet-scale image and text data, has shown impressive robustness benefits on ImageNet variants such as ImageNetV2. Similarly, pre-training methods such as BERT (Devlin et al., 2019) have improved robustness in NLP — for example, multilingual pre-training significantly improves performance on unseen languages (Liu et al., 2020). While other robustness methods that use unlabeled data such as self-training (Prabhu et al., 2021; Sohn et al., 2020; Berthelot et al., 2021) and domain adversarial training (Shu et al., 2018; Ganin et al., 2016) jointly optimize a labeled and unlabeled objective, pre-training is particularly suitable for large unlabeled datasets since we can generically pre-train once on unlabeled data and then specialize the model to downstream labeled data via fine-tuning. How does pre-training improve robustness to distribution shift when the pre-training method is not tailored to any downstream task?

In this paper, we run controlled experiments to understand the robustness benefits of a popular pre-training method, contrastive pre-training (Chen et al., 2020; He et al., 2020; Caron et al., 2020), even without large-scale unlabeled data. We directly apply contrastive pre-training to unsupervised visual domain adaptation (UDA), where unlabeled data from a shifted target domain is used to adapt models trained on labeled source dataset. We compare contrastive pre-training (SwAV (Caron et al., 2020)) to UDA methods such as SENTRY (Prabhu et al., 2021) (self-training) and DIRT-T (Shu et al., 2018) (domain adversarial training) where all methods use the same unlabeled data (source and target). We find that SwAV achieves comparable or better results on the DomainNet, BREEDS Living-17, and BREEDS Entity-30 benchmarks. Furthermore, we show that pre-training once on unlabeled data from many domains further improves the average target accuracy over the consituent source-target pairs, showing that extra unlabeled data improves contrastive pre-training for domain adaptation.

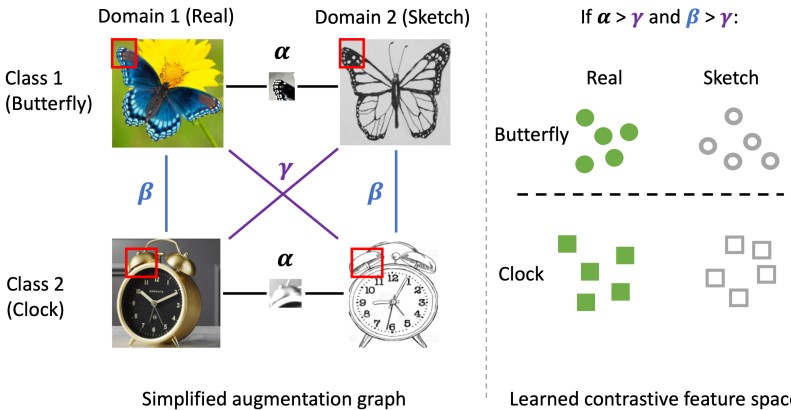

Figure 1: **(Left)** Edges between (class, domain) pairs indicate how "connected" they are by augmentations (e.g., cropping, colorization). Small crop sizes in contrastive learning can increase the connectivity between disparate domains. For example, $\alpha$ denotes the probability that augmentations connect examples of the same class across different domains. **(Right)** When $\alpha$ and $\beta$, which measures the connectivity between images of the same class or same domain respectively, are larger than $\gamma$ (different domain and different class connecivity), contrastive pre-training produces a feature space where training on labeled data from the source domain (Real — green, filled) achieves high accuracy on the target domain (Sketch — gray, hollow).

Given the strong performance of contrastive pre-training, we seek to understand why contrastive pre-training works for domain adaptation. Domain adversarial training and self-supervision algorithms are commonly based on the intuition that domains get merged in feature space (Tzeng et al., 2014; Ganin et al., 2016; Tzeng et al., 2017; Sun et al., 2019; Ben-David et al., 2010; Shu et al., 2018; Wang et al., 2021; Thota & Leontidis, 2021). This is support by theoretical notions such as $\mathcal{H}\Delta\mathcal{H}$-divergence, where domains that can be discriminated well and results in worse generalization bounds for target accuracy (Ben-David et al., 2010). A natural hypothesis is that contrastive pre-training brings domains together by bringing the representations of strong augmentations of the same image together (Sun et al., 2019).

Surprisingly, we find that the source and target domains are still far apart in the feature space learned by contrastive pre-training. A domain classifier trained on the pre-trained representations of strongly augmented images from the sketch and painting domains in DomainNet has 94% accuracy, which is even higher than the accuracy of a domain classifier on the original (augmented) input space (75%).

We study this phenomenon from the lens of the recently proposed augmentation graph (HaoChen et al., 2021), which forms connections between examples if they have similar augmentations (e.g., a small crop + grayscale can connect sketch and real photos, Figure 1). These connections are central to the contrastive objective, which seeks to bring neighbors in the graph closer together while pushing apart non-neighbors. From both simulations and real datasets, we define a critical connectivity quantity which seems to govern whether domain adaptation will be successful (Figure 1). In particular, the connection strength between the same classes across different domains should be relatively greater than the connection between different classes across different domains. Importantly, the absolute connectivity between domains can be small and the domains can be far apart in input space. This suggests that contrastive learning circumvents the separation of domains through the connectivity structure of the (general-purpose) augmentations used.

Using our understanding, we ablate contrastive learning by removing subsets of unlabeled data that contribute most to connecting the two domains. We do this by training a domain classifier to distinguish between the source and target and removing examples which are most uncertain under the classifier. This consistently worsens the target accuracy in comparison to randomly removing the same number of data points (by 2–3% on DomainNet), showing the importance of domain connectivity for adaptation.

## 2 SETUP

We consider a classification problem from an input space $\overline{\mathcal{X}} \subseteq \mathbb{R}^d$ to a label space $\mathcal{Y} = [K]$.

**Data.** Let $P_S$ and $P_T$ denote the source and target distributions of $(\bar{x}, y)$ pairs, respectively, where $\bar{x} \in \overline{\mathcal{X}}$ and $y \in \mathcal{Y}$. The labeled source dataset $S$ consists of $n_S$ input-output pairs from $P_S$. We also have access to an unlabeled target dataset $T$ with $n_T$ unlabeled inputs from $P_T$, and we assume the label sets for the source and target are identical. In some cases, we also consider an additional related dataset $R$ with $n_R$ unlabeled inputs from a related distribution $R$.

**Model and metrics.** The goal is to learn a classifier $h : \overline{\mathcal{X}} \to \mathcal{Y}$ with low classification error on the target distribution $\mathcal{L}_{0-1}(h) = \mathbb{E}_{\bar{x}, y \sim P_T}[\mathbf{1}[h(\bar{x}) \neq y]]$. The pre-training algorithms we consider consist of two steps: first, we train an encoder function $f : \mathcal{X} \to \mathcal{Z}$ into a representation space $\mathcal{Z} \subseteq \mathbb{R}^{d'}$ (where $d'$ denotes the representation dimension), then we train a classification head $g : \mathcal{Z} \to \mathcal{Y}$. The final classifier $h = g \circ f$ composes the encoder with the classification head.

### 2.1 CONTRASTIVE LEARNING

Contrastive learning (He et al., 2020; Chen et al., 2020; Caron et al., 2020) trains representations to be similar between augmented views of the same example and dissimilar between augmented views of random examples. Example image augmentations include random cropping, color jitter, and Gaussian blur (Chen et al., 2020). The basic assumption is that the semantic content of the data is invariant under the augmentations.

Though there are several variants of contrastive learning, we focus on the following objective from HaoChen et al. (2021) for our conceptual understanding. Let $\mathcal{X}$ denote the "augmented" space containing all data-augmented views of natural examples (e.g., all random crops of natural ImageNet images). Data augmentation is defined as a random function $\mathcal{A} : \overline{\mathcal{X}} \to \mathcal{X}$, where $\mathbb{P}[\mathcal{A}(\overline{x}) = x]$ denotes the probability that $\overline{x} \in \overline{\mathcal{X}}$ augments to $x \in \mathcal{X}$. A pair of augmented views is called a positive pair and denoted $x, x^+$ if a natural point $\overline{x}$ is first drawn and then $x, x^+$ are the results of two calls to $\mathcal{A}(\overline{x})$. On the other hand, a negative pair $x, x^-$ arises from two natural data points $\overline{x}, \overline{x}'$ when $x = \mathcal{A}(\overline{x})$, $x^- = \mathcal{A}(\overline{x}')$. Minimizing the following loss function simultaneously brings together the representations of positive pairs $x, x^+$ and repels the representations of negative pairs $x, x^-$:

$$\mathcal{L}(f) = -2 \cdot \mathbb{E}_{x, x^+}\left[ f(x)^\top f(x^+) \right] + \mathbb{E}_{x, x^-}\left[ \left( f(x)^\top f(x^-) \right)^2 \right]. \tag{1}$$

Here the first expectations are taken over the marginal distribution over $\overline{\mathcal{X}}$ and the randomness of $\mathcal{A}$.

### 2.2 CONSTRASTIVE PRE-TRAINING FOR DOMAIN ADAPTATION

Let $D$ denote the unlabeled input data used for contrastive pre-training (e.g., some combination of the source unlabeled data $S$, target unlabeled data $T$, and/or additional unlabeled data $R$). We use the following simple algorithm to apply contrastive pre-training to domain adaptation:

1. (Pre-train) Learn an encoder function $f : \mathcal{X} \to \mathcal{Z}$ via contrastive learning on $D$.

2. (Fine-tune) On labeled source data $S$, train a classifier $h \triangleq g \circ f : \mathcal{X} \to \mathcal{Y}$ as the composition of the pre-trained encoder $f$ and a classifier head $g$. In this paper, we take "fine-tuning" to mean updating both the parameters of $f$ and $g$.

## 3 CONTRASTIVE PRE-TRAINING IS A STRONG DOMAIN ADAPTATION METHOD

We evaluate contrastive learning and relevant baselines on 4 benchmark vision datasets and observe that contrastive learning achieves comparable or better performance in all cases.

### 3.1 DATASETS

**BREEDS (Santurkar et al., 2020).** BREEDS is a subpopulation shift benchmark derived from ImageNet by constructing a hierarchical tree structure of classes from WordNet. Nodes at a specified depth of the tree become the labels for the classification task, and descendant nodes are treated as subpopulations that can be randomly partitioned into source and target domains.

**DomainNet (Peng et al., 2019).** DomainNet is a large unsupervised domain adaptation task, consisting of approximately 600,000 images and 345 classes in 6 domains. For our experiments we utilize the same filtered version of DomainNet from Prabhu et al. (2021), which uses 40 of the 345 classes and the sketch, painting, photograph, and clipart domains.

| Source | Real | | | Sketch | | | Painting | | | Clipart | | | Avg. |
|---|---|---|---|---|---|---|---|---|---|---|---|---|---|
| Target | Sketch | Painting | Clipart | Real | Painting | Clipart | Real | Sketch | Clipart | Real | Sketch | Painting | |
| ERM | 25.67 | 37.09 | 44.80 | 25.72 | 18.59 | 31.87 | 29.13 | 14.25 | 20.48 | 19.34 | 12.71 | 10.79 | 24.20 |
| ERM (SA) | 42.76 | 47.26 | 48.26 | 26.74 | 18.56 | 37.56 | 40.32 | 32.01 | 29.82 | 26.89 | 22.13 | 14.30 | 32.22 |
| SENTRY | 41.18 | 51.66 | **59.65** | 56.73 | **42.55** | 55.01 | 30.13 | 13.33 | 20.97 | 10.52 | 5.33 | 6.60 | 32.81 |
| DANN | 44.97 | 48.88 | 57.36 | 46.26 | 39.60 | 52.28 | 47.77 | 32.22 | 35.64 | 32.98 | 25.46 | 15.67 | 39.92 |
| DANN (SA) | **52.52** | 50.91 | 57.42 | 45.32 | 37.02 | **56.74** | 48.46 | **49.39** | **40.22** | 33.97 | 32.34 | 23.58 | 43.99 |
| SwAV | 43.76 | **54.55** | 53.27 | 55.48 | 34.99 | 40.59 | **67.08** | 39.64 | 32.98 | **57.13** | **34.30** | **25.09** | **44.91** |
| SwAV+ | 44.64 | 57.27 | 54.20 | 58.10 | 46.75 | 53.46 | 69.03 | 48.68 | 41.33 | 59.38 | 46.22 | 41.66 | 51.73 |

| | ERM | SENTRY | DANN | SwAV (S) | SwAV (T) | SwAV (S+T) | SwAV+ |
|---|---|---|---|---|---|---|---|
| Living-17 | 59.00 | 68.76 | 66.88 | 61.89 | 68.53 | **73.59** | 81.82 |
| Entity-30 | 50.97 | **62.80** | 54.62 | 52.33 | 60.33 | 62.03 | 65.90 |

| ERM | Dirt-T | SimCLR (T) |
|---|---|---|
| 63.6 | 75.3 | **76.1** |

Table 1: **(Top)** Test accuracy (%) of ERM with standard and SwAV augmentations, SENTRY, DANN with standard and SwAV augmentations, SwAV, and SwAV + extra (abbreviated to SwAV+) on all domain pairs of DomainNet. Here, SwAV is run on source + target, and SwAV+ is separated from the other algorithms as it uses additional data and is therefore not directly comparable. **(Middle)** Test accuracy (%) of ERM, SENTRY, DANN, SwAV (with various splits), and SwAV+ on BREEDS. **(Bottom)** Test accuracy (%) of ERM, Dirt-T, and SimCLR on STL-10 → CIFAR-10. ERM and Dirt-T numbers are as reported from Shu et al. (2018).

**STL-10 → CIFAR-10.** STL-10 and CIFAR-10 are two classic image classification datasets, each consisting of 10 classes (Coates et al., 2011; Krizhevsky, 2009). Each dataset has a class label that is not present in the other, and hence we follow the procedure of French et al. (2018) and filter out the examples in the non-overlapping classes, resulting in a 9-class classification problem.

## 3.2 METHODS

**Baselines.** We run empirical risk minimization (ERM) on the source dataset for each task, i.e., a "source-only" baseline. In addition, for BREEDS and DomainNet we use these ERM models as weight initialization for SENTRY, a self-training algorithm for domain adaptation that achieved state of the art results on several domains of DomainNet when initialized with ImageNet-pretrained models (Prabhu et al., 2021). For STL → CIFAR, we use Dirt-T (Shu et al., 2018), the current state of the art algorithm for this task.

**Contrastive pre-training.** For BREEDS and DomainNet we use SwAV (Caron et al., 2020), a contrastive learning algorithm that achieved state-of-the-art linear evaluation accuracy on ImageNet. SwAV contrasts examples in feature space rather than input space and uses a novel multi-crop data augmentation strategy using several crops of different sizes. For the lower-resolution task STL → CIFAR we use SimCLR (Chen et al., 2020), which uses only 2 crops and therefore scales straightforwardly to smaller input dimensions.

## 3.3 RESULTS

**Main comparison.** For all 4 tasks, contrastive pre-training is competitive when compared to ERM and SOTA unsupervised domain adaptation baselines. On DomainNet (top of Table 1), contrastive pre-training on source and target unlabeled data achieves higher target accuracy than all joint-training baselines both on 5 of 12 domain pairs and on average over all pairs (44.91% vs. the second best, 43.99%). On Living-17 (middle of Table 1), contrastive learning on source and target unlabeled data improves over SENTRY over 4%, while on Entity-30, contrastive learning (source + target) is within 0.3% of the target accuracy of SENTRY. On STL→CIFAR, SimCLR on the target unlabeled data achieves 0.8% higher target accuracy than Dirt-T.

**Extra unlabeled data from related domains.** We also consider adding extra unlabeled data from other (related) domains in DomainNet, Living-17, and Entity-30. In particular, we can pre-train once on an unlabeled dataset that consists of a superset of the domains we want to adapt to, and fine-tune on the available source data. While this is not a fair comparison to other domain adaptation methods, the ability to scale to large unlabeled datasets is a natural advantage of pre-training.

| | Feature space learned by | Different class, same domain, source | Different class, same domain, target | Same class, different domain | Different class, different domain |
|---|---|---|---|---|---|
| Living-17 | Input space | 20.94 | 21.91 | 36.58 | 20.00 |
| | SwAV | 1.85 | 2.26 | 12.67 | 1.26 |
| DomainNet | Input space | 32.88 | – | 27.36 | 25.12 |
| | ERM | 15.16 | 16.85 | 18.34 | 10.69 |
| | DANN | 10.59 | 11.96 | 16.59 | 7.30 |
| | SwAV | 7.03 | – | 7.54 | 2.46 |

Table 2: Average separation error of Living-17 and DomainNet class-domain pairs. On both datasets, the classes become very distinguishable in the feature space learned by contrastive learning and on DomainNet, the domains additionally remain far apart in the pre-trained feature space. Classifiers were trained on augmented images to distinguish class-domain pairs as a proxy for connectivity (higher separation errors suggest greater connectivity).

On DomainNet, we consider contrastive pre-training with SwAV on unlabeled data from all the domains at once, which we denote as SwAV + extra. The top Table 1 shows that adding extra unlabeled data improves the target accuracy of contrastive pre-training on all pairs and improves the average by almost 6% (44.91% to 50.75%). On Living-17 and Entity-30 (middle of Table 1), we additionally consider contrastive pre-training with SwAV with 1) only source, 2) only target, and 3) with all of ImageNet as the unlabeled data. We find that increasing the amount of unlabeled data steadily increases the target accuracy, and pre-training on all of ImageNet improves over source + target by 8% and 4% respectively for Living-17 and Entity-30.

**Ease of hyperparmeter tuning.** The SENTRY self-training baseline requires extensive hyper-parameter tuning (e.g., finding optimal balancing coefficients for the joint losses, or learning rate schedules) for *every* transfer task and is difficult with access only to source labels. In contrast to the baseline methods, for contrastive pre-training we did not need to sweep over hyperparameters or use target labels for model selection. We used nearly identical hyperparmeters for both BREEDS datasets (Living-17 and Entity-30) and all DomainNet pairs (details in Appendix A).

**No explicit knowledge of distribution shift.** We note that SENTRY uses explicit knowledge of label imbalance within its algorithm, on top of being designed for the domain adaptation task of adapting from source to target. In contrast, we used off-the-shelf contrastive pre-training, which was designed for general representation learning, with no additional information about distribution shift.

## 4 CONNECTIVITY MODEL FOR DOMAIN ADAPTATION

How does contrastive pre-training improve target accuracy with unlabeled data? Conventional intuitions and theory for domain adaptation are based on reducing the distance between the source and target domains (Ben-David et al., 2010; Ganin et al., 2016).However, we see empirically that contrastive pre-training keep the domains very separated in representation space. Towards building an understanding for how contrastive pre-training connects domains without merging them, we extend a conceptual framework based on connectivity via data augmentations (HaoChen et al., 2021) for distribution shifts. We define a crucial measure of *domain connectivity* that relates properties of the data distribution and contrastive augmentations to the success of contrastive learning for domain shifts on a simulated data model, DomainNet, Living-17, and Entity-30.

### 4.1 CONTRASTIVE PRE-TRAINING KEEPS DOMAINS APART

Contrary to common intuitions for domain adaptation methods, we find that contrastive learning does not bring the domains together in feature space. Table 2 shows the average test error of classifiers trained to discriminate between examples from the same class but different domains on DomainNet. We train and test the classifiers on images using the same augmentations as SwAV and find that contrastive pre-training learns features that are actually more discriminable between domains than in the original input space (7.5% vs. 27.3% error on average over all pairs). These estimates for the baselines and for individual pairs are provided in table 3.

Next, we measure the separability in feature space and input space between classes within a single domain (e.g., 3% vs. 24% error in the real domain, table 3). We note that the *separability between domains in feature space is comparable to the separability between classes*, suggesting that contrastive learning may cluster both domains and classes in separate axes.

### 4.2 USING CONNECTIVITY TO CHARACTERIZE DOMAIN SHIFT

In this section, we set up our conceptual model of connectivity used to study contrastive learning.

**Connectivity via augmentations in domain adaptation.** HaoChen et al. (2021) proposed viewing contrastive learning as operating on an augmentation graph. The augmented inputs from $\mathcal{X}$ constitute the vertices and the edge weight between $x, x'$ represents how likely they are to be generated from the same natural example (i.e., how likely they are to be chosen as a positive example during contrastive pre-training). HaoChen et al. (2021) show that when examples from different classes $\overline{x}, \overline{x}' \in \overline{\mathcal{X}}$ are sparsely connected via the augmentation graph (i.e., rarely augment into the same augmented example), and the connectivity within a class is strong, the contrastive pre-training leads to features that obtain good accuracy on the downstream task.

**Extension to domain adaptation.** Motivated by the augmentation graph framework, we define a notion of connectivity for domain adaptation on a graph where *class-domain pairs* are vertices. We denote the set of examples belonging to class $c$ and domain $d$ (which we refer to as the class-domain pair $(c, d)$) as $P_{(c,d)}$. We formalize this notion in the following definition:

**Definition 1** (Connectivity). *The connectivity between two class-domain pairs $(i, d)$ and $(j, d')$ is*

$$\text{conn}((i,d),(j,d')) \triangleq \mathbb{P}[\mathcal{A}(\overline{x}) = \mathcal{A}(\overline{x}')],$$

*where $\overline{x} \sim P_{(i,d)}$ and $\overline{x}' \sim P_{(j,d')}$.*

Intuitively, the connectivity between two class-domain pairs is the probability that two randomly sampled points from each class-domain pair are augmented to the same point.

Definition 1 is defined for two arbitrary class-domain pairs, which then allows us to build further definitions of connectivity involving multiple class-domain pairs. Such connectivity definitions would allow us to more rigorously describe (for instance) our observation in the DomainNet dataset that points of the same class but different domains are more discriminable than points in different classes within the same domain. We define these instances of connectivity using Definition 1 below:

**Definition 2** ($(\alpha, \beta, \gamma)$ - connectivity). *A source-target pair of distributions $S$ and $T$ over $\overline{\mathcal{X}}$ (with domain labels $d_S$, $d_T$, respectively) satisfies $(\alpha, \beta, \gamma)$ - connectivity if*

1. $\mathbb{E}_{i \in [K]}[\text{conn}((i, d_S), (i, d_T))] = \alpha$. *(same-class-different-domain)*

2. $\mathbb{E}_{i,j \in [K]}[\text{conn}((i, d), (j, d))] = \beta$. *(different-class-same-domain, $d \in \{d_S, d_T\}$ )*

3. $\mathbb{E}_{i,j \in [K]}[\text{conn}((i, d_S), (j, d_T))] = \gamma$. *(different-class-different-domain)*

### 4.3 ILLUSTRATIVE TOY EXAMPLE: BINARY CLASSIFICATION

As an illustrative example, consider the simple case of binary classification where each class-domain pair consists of a single data point. This toy setting pinpoints a simple connectivity condition such that linear classification on the pre-trained features results in good target accuracy.

**Setting (Figure 2).** Assume a uniform marginal distribution over the natural inputs, and suppose data augmentation transforms each data point directly into another with a certain probability, so that the entire augmentation graph consists of the 4 points (2 classes, 2 domains) each identified by their class and domain. Formally, the example $(i, d)$ (of class $i$ and domain $d$) has augmentation given by

$$\mathcal{A}((i,d)) = \begin{cases} (i,d) & \text{with probability } \zeta \\ (i,d') & \text{with probability } (\alpha/2\zeta) \\ (j,d) & \text{with probability } (\beta/2\zeta) \\ (j,d') & \text{with probability } (\gamma/2\zeta) \end{cases}$$

where $j \neq i$, $d' \neq d$. We also assume $\zeta > \max\{\alpha, \beta, \gamma\}$ (i.e., data augmentation is most likely to map each data point to itself) and appropriately constrain $\alpha + \beta + \gamma = 2\zeta(1 - \zeta)$ so that the probabilities sum to 1. Note that the specific transition probabilites are selected so that the distribution shift satisfies $(\alpha, \beta, \gamma)$ - connectivity. We also number the points $1 - 4$ in the following order: class 1 (source), class 1 (target), class 2 (source), class 2 (target).

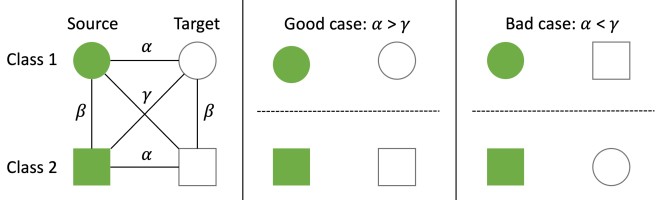
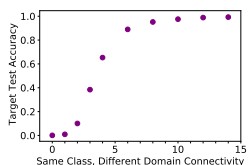

Figure 2: **(Left)** Illustrative toy example with 2 domains and 2 classes, where each class-domain pair is a single node in the graph. Edge weights denote connectivities (probability of two class-domain pairs augmenting into the same point). **(Middle)** When $\alpha$ (same-class-different-domain connectivity) is greater than $\gamma$ (connectivity across different domains and classes), the domain representation are oriented so that linear classification on the source classifies the target accurately. **(Right)** When $\alpha < \gamma$, no linear classifier can label both the source and target accurately.

Figure 3: Scatter plot of random graph simulations, where $\zeta = 16, \beta = 4$, and $\gamma = 2$. Shown on the x-axis is $\alpha$ and nontrivial target accuracy is not possible until $\alpha > \gamma$. Target accuracy also increases with the ratio $\alpha/\beta$.

**Pre-training.** HaoChen et al. (2021) show that minimizing the spectral contrastive loss (given in Eq. 1) is equivalent to computing the spectral decomposition of the adjacency matrix of the population augmentation graph. The leading eigenvectors of the resulting decomposition are then used as the learned representations, so that the representation of example $i$ is the concatenation of the $i^{th}$ components of those eigenvectors. Because the population here is finite, we compute the spectral decomposition algebraically and express the learned representations as a function of connectivity. Agnostic of the ordering of the relevant parameters, the *unordered* eigenvectors of this toy example are:[1]

$$v_1 = [\,1\,1\,1\,1\,]^\top, \quad v_2 = [\,1\,-1\,1\,-1\,]^\top, \quad v_3 = [\,1\,1\,-1\,-1\,]^\top, \quad v_4 = [\,1\,-1\,-1\,1\,]^\top$$

with associated eigenvalues

$$\lambda_1 = \zeta + \beta + \alpha + \gamma, \quad \lambda_2 = \zeta + \beta - \alpha - \gamma, \quad \lambda_3 = \zeta - \beta + \alpha - \gamma, \quad \lambda_4 = \zeta - \beta - \alpha + \gamma$$

We discard the eigenvector $v_1$, as it assigns the same feature to every point, and use the top 2 remaining eigenvectors as the learned features. For example, if $v_2$ and $v_3$ are the top remaining eigenvectors, then the features for data point 2 would be the $2^{nd}$ component of $v_2$ and $v_3$ (namely, $(-1,1)$).

**Fine-tuning.** We consider learning a linear classifier upon the pre-trained features. The middle panel of Figure 2 illustrates the fortunate case, in which $\alpha > \gamma$ and $\beta > \gamma$ (but possibly $\alpha \leq \beta$). In this case, $\lambda_2 > \lambda_4$ and $\lambda_3 > \lambda_4$ and therefore the learned features use $v_2$ and $v_3$ and are $(1,1),(1,-1)$ for class 1 and $(-1,1),(-1,-1)$ for class 2. Then, a linear classifier trained on the source uses only the class information, which is contained in the second index, and labels the target accurately.

Note that the source classifier labels the target correctly if and only if it uses solely the features given by $v_3$ for prediction. Two possible failure modes occur when $\alpha$ is smallest (and therefore $v_3$ is discarded) and when $\beta$ is smallest (and therefore the source classifier can use features from either $v_3$ or $v_4$).

Importantly in this toy setting, whether or not the features of the target examples are oriented correctly depends only on whether $\alpha$ and $\beta$ are both $> \gamma$. Consequently, the classes within each domain can be very connected–more so than across domains–and target accuracy can be high. To illustrate the importance of this connectivity ratio on a more fine-grained setting, we run simulations and observe sharp thresholding behavior when $\gamma$ surpasses $\alpha$ (Figure 3, simulation details in Appendix C).

## 4.4 Connectivity Governs the Success of Pre-training for Domain Adaptation

**Connectivity on benchmark datasets.** We estimate the input space domain connectivity on DomainNet, Living-17, and Entity-30 by training classifiers between various pairs of class-domain pairs (test errors provided in Table 2, full details provided in Section B). We find that all the datasets satisfy $\alpha > \gamma$–the necessary condition for the good case in the toy example–but $\alpha < \beta$ for DomainNet while $\alpha > \beta$ for BREEDS. Intuitively, on natural distribution shifts $\alpha$ is likely to be higher than $\gamma$, since the data augmentation should be more likely to connect class-domain pairs where only the domain differs than to bridge differences across both the domain and the class.

---

[1]We leave the eigenvectors unnormalized for ease of exposition; the normalized variants would multiply the entire vector by 1/2.

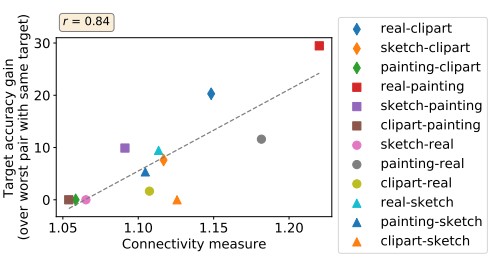

Figure 4: Plot of our measure of connectivity ratios against target accuracy of contrastive pre-training on DomainNet. Our quantity highly correlates $r = 0.84$ with target accuracy.

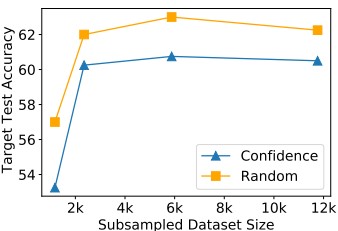

Figure 5: Ablation of connectivity by removing unlabeled data points near the margin of a domain classifier. Removing these points from the pre-training dataset reduces target accuracy more than removing a random subset of the same size.

**Domain and class classifiers are nearly orthogonal.** We check the extent to which the domain and class information are separated by training 3 linear classifiers on the pre-trained representation space: 1) Source classifier (which discriminates classes in source) 2) Target classifier (which discriminates classes in target)[2] 3) Domain classifier (discriminates between source or target domains). We compare the linear weights of these classifiers by computing the cosine similarity. We find that

1. The source and target classifiers are very similar; on average over the classes, the cosine similarity of linear weights for the same class from the source and target classifiers is high (around 0.20 for DomainNet, 0.34 for Living-17).

2. The domain classifier is nearly orthogonal to the source and target classifiers. The average cosine similarity between the weights of any class from the source or target classifiers and the weights of the domain classifier is 0.01.

This suggests that the class information and domain information are contained in different directions of the feature space, and gives an explanation for how both classes and domains can be easily separated in feature space while maintaining the ability to generalize from source to target. Learning to discriminate classes results in a classifier with weights that are nearly orthogonal to the weights for domain classification, suggesting a natural amount of 'domain agnosticity" despite not removing domain information from the features. Detailed results on individual pairs are provided in Appendix D.

**Connectivity ratios correlate with target accuracy.** We construct a scalar connectivity quantity and find that it correlates with target accuracy of contrastive learning on DomainNet. In the simple example, contrastive pre-training for domain adaptation works when $\alpha > \gamma$ and $\beta > \gamma$. The first condition ($\alpha > \gamma$) is relatively more important since if $\beta$ (connectivity between different classes in the same domain) is very high, then the classes within a domain are hard to discriminate and the classification task itself will be hard. Thus, we define the quantity

$$(\alpha/\gamma) \cdot (\beta/\gamma)^{\epsilon} \tag{2}$$

where we take $\epsilon = 0.1$ to "discount" the importance of the second ratio. We multiply the two ratios to express the logical "and". The larger this quantity is, the better the target accuracy should be.

Figure 4 shows a plot of the quantity with target accuracy of contrastive learning on DomainNet pairs. Because different target domains can have different intrinsic errors, we "debias" the target accuracy of each source-target pair by subtracting the worst accuracy over all pairs with the same target domain. We find a strong correlation between the connectivity and target accuracy (Pearson $r$ of 0.87 for S+T pre-training) (Figure 4).

**Ablating connectivity.** We find that the connectivity between examples of the same class but different domain is also very important, as those examples "bridge" the domains. We verify this intuition by systematically ablating the connectivity on a subset of Living-17 by removing those examples that contribute most to domain connectivity. Specifically, for each class we trained discriminators between the source and target examples of that class and removed the $n$ pre-training data points from each

---

[2]Although this cannot be done in practice, we only use the target labels here for exploratory analysis;

class-domain pair on which the discriminator was least confident; these examples can be seen as the ones contributing most to domain connectivity. For subsample sizes of $\{1176,2352,5880,11760\}$ we consistently reduce target accuracy as compared to random subset removal (Figure 5; confidence subsampling increases error relatively by 6% on average compared to random).

## 5 RELATED WORK

**Domain adaptation.**    Ben-David et al. (2010) prove generalization bounds in the distribution shift setting. Their bounds rely on a notion of distance between two data distributions called $\mathcal{H}\Delta\mathcal{H}$ divergence being small, which many methods such as domain adversarial training are inspired by (Ganin et al., 2016; Tzeng et al., 2014; 2017). Our conceptual model based around connectivity provides a way to explain how contrastive pre-training can work for domain adaptation even when the $\mathcal{H}\Delta\mathcal{H}$ divergence between domains is large.

**Domain adaptation with self-supervision.**    Prior works have explored the application of self-supervision to domain adaptation. Sun et al. (2019) propose optimizing on hand-crafted self-supervised tasks such as predicting the angle of rotation of a rotated image. These tasks are jointly optimized on the source and target data along with the labeled source loss. As part of their comparisons, they found that first pre-training on target (or source and target) data and then fine-tuning on source data separately (instead of jointly) was highly non-competitive with their other methods. In contrast, we find that pre-training with a general, off-the-shelf contrastive learning objective is competitive with other state-of-the-art domain adaptation techniques.

**Contrastive objective for domain adaptation.**    Contrastive objectives have also been applied to domain adaptation problems. Wang et al. (2021) jointly train the contrastive loss on source and target data. These methods rely on explicit anchor pairs that encourage domain alignment (Wang et al., 2021), or still employ some form of explicit domain alignment (Thota & Leontidis, 2021). We consider pre-training instead of joint training and do not use any domain knowledge about the distribution shift problem.

**Self-training.**    Berthelot et al. (2021) use self-training and consistency regularization with a distribution alignment method, but require estimating the target label distribution and tuning a confidence thresholding parameter (tuned using *target test labels*). Cai et al. (2021) give a label propagation analysis for self-training in domain adaptation.They have only one parameter which governs both the different-class-same-domain and different-class-different-domain connections, requiring both to be small. In our framework, we allow the different-class-same-domain connectivity to not be small, in accordance with our observation that domains can be well separated in contrastive feature space. They give experiments on self-training when operating on a contrastive pre-trained representation space, but their focus is on self-training while the pre-trained representations are an empirical detail. One inflexible characteristic of standard self-training is that it relies on the unlabeled data having the same classes as the labeled data (in order to pseudolabel unlabeled points), which may not allow them to utilize diverse sources of extra unlabled data like many pre-training methods can.

## 6 CONCLUSION

In this work we focused on evaluating and understanding contrastive pre-training under domain adaptation. Pre-training has the advantage of amortizing the cost of processing a large unlabeled dataset into a pre-training step that is done only once. While pre-training loses the advantage of seeing labeled data (and thus the downstream task) to adapt its usage of unlabeled data, we find that contrastive pre-training is still competitive with other domain adaptation algorithms. We hope that our connectivity model can give insights to improve pre-training data selection, develop better augmentations for pre-training with distribution shift in mind, and improve fine-tuning methods that exploit the properties of the pre-trained feature space.

## ETHICS

Our experiments use standard benchmark vision datasets in domain adaptation, which is publicly available data. In general, however, large-scale unsupervised learning can involve data scraped from the internet which may lead to privacy concerns. Care should be taken when curating datasets for large scale contrastive pre-training in practice.

## Reproducibility

We detail the hyperparameters and experimental details in the paper and appendix. We have uploaded a zipped code archive and will publish the code on Github upon acceptance.

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

## A  ADDITIONAL EXPERIMENTAL DETAILS

### A.1  DATASETS

**BREEDS.**  We use the dataset creation functions defined in the `robustness` Python library to generate the Living-17 and Entity-30 tasks from the original ImageNet dataset (Engstrom et al., 2019; Russakovsky et al., 2015). The Living-17 dataset is an animal classification task which consists of nodes in the subtree rooted at the "living thing" node in the WordNet hierarchy. An example of a label is "ape" with subpopulations of gibbons, orangutans, gorillas, and chimpanzees. The Entity-30 dataset is a much more general task, incorporating nodes in the "entity" subtree. Labels include "building" and "home appliance". The trailing number in the dataset name is the total number of classes in that dataset.

**DomainNet.**  We use the official SENTRY repo at `https://github.com/virajprabhu/SENTRY`, which filters the original DomainNet dataset automatically as described in Section 3. This refinement is done to eliminate much of the noise present in the original DomainNet dataset (Tan et al., 2020).

**STL → CIFAR.**  CIFAR-10 consists of $32 \times 32$ images from the former TinyImages dataset, and STL-10 is derived from ImageNet and contains images with resolution $96 \times 96$. We resize the STL-10 images to $32 \times 32$ to match the resolution of CIFAR10. The two non-overlapping classes ("monkey" in CIFAR-10 and "frog" in STL10) are removed from both datasets before training.

### A.2  BASELINES AND HYPERPARAMETER TUNING

**ERM (source-only).**

- BREEDS: We use the same hyperparameters as Santurkar et al. (2020). For Entity30, we train for 300 epochs and divided the learning rate by 10 every 100 epochs. On Living17, we train for 450 epochs and divided the learning rate by 10 every 150 epochs. For both datasets we use a ResNet50 with a learning rate of $0.1$, a weight decay of $10^{-4}$, and a batch size of 128.

- DomainNet: The SENTRY algorithm runs ERM with class balancing (starting with ImageNet-pretrained initialization) prior to beginning entropy minimization, and therefore the SENTRY repository contains an ERM implementation and hyperparameters for DomainNet. We run ERM for 150 epochs and multiply the initial learning rate by 10x, keeping all other hyperparameters constant.

- STL10 → CIFAR10: We report the exact results from Shu et al. (2018).

**DIRT-T.**  DIRT-T (Shu et al., 2018) is a domain adaptation method that addresses two flaws of domain adversarial neural networks (Ganin et al., 2016): 1) distribution matching is a weak constraint, and 2) in some domain adaptation settings there does not exist a good joint classifier on both source and target. The authors address the first shortcoming by adding a conditional entropy regularization term so that the model's decision boundaries do not overlap high-density regions of data. This is inspired by the *cluster assumption*, which states that the input space is divided into well-separated clusters, one for each class in the label space. The lack of a good classifier on both source and target is then addressed via self-training on the unlabeled target data. We report the STL10 → CIFAR10 results from Shu et al. (2018).

**SENTRY.**  For each pair of domains on DomainNet we conduct a hyperparameter search through $\lambda_{\text{src}} \in \{0.5, 1.0, 1.5\}$ (the weight on the supervised classification loss) and learning rates $\in \{0.01, 0.001\}$ and report the model results with the highest source evaluation accuracy.

On BREEDS we keep the same default hyperparameters from the SENTRY repo and search over 3 learning rates $\{0.001, 0.01, 0.1\}$, choosing the model with the highest target evaluation accuracy to report.

SENTRY underperforms the ERM models when the source domain is clipart images (from DomainNet), which could be the result of not using ImageNet-pretrained classifiers for initialization as Prabhu et al. (2021) do. Therefore, as the joint-training baseline for DomainNet we take the maximum of ERM and SENTRY.

### A.3  SIMCLR

We use the official SimCLR repository from Google for our experiments (`https://github.com/google-research/simclr`). We use a ResNet18 with a batch size of 256, a learning rate of 0.2, and weight decay $10^{-4}$. The projection head has two layers and an output

dimension of 64. We pre-train the model for 400 epochs with square-root learning rate scaling and we train the linear probe for 100 epochs on batches of size 512 and a learning rate of 0.1.

### A.4 SwAV

We use the public SwAV repository available at `https://github.com/facebookresearch/swav` and kept almost all of the hyperparameters provided by the original paper for 400 epoch, 256-batch-size training on ImageNet. However, we used a batch size of 512 and additionally made the following changes based on the Github issues answered by the original authors:

1. We avoid using a queue on DomainNet and the subsampled variant of Living-17 in order to stabilize training. For pre-training on the full Living-17 and Entity-30 datasets, we introduce the queue at epoch 60.

2. We set the number of prototypes to be 10 times the number of classes (170, 300, and 400 for Living-17, Entity-30 and DomainNet, respectively).

3. We set $\epsilon = 0.03$.

4. We set the base learning rate to 0.6, following a linear scaling rule based on batch size.

We always fine-tuned from the final iterate of SwAV pre-training (400 epochs).

On DomainNet, we fine-tuned SwAV models for 50 epochs using the ERM implementation in the SEN-TRY repository (without running any joint-training algorithm), keeping all hyperparameters other than the number of epochs constant. We report the target test accuracy of the final model (after 50 epochs).

On BREEDS Living-17, we fine-tuned SwAV models for 100 epochs with a cosine learning rate schedule without restarts. We use SGD with initial learning rate 0.1 for the classifier head and 0.01 for the encoder, momentum 0.9, and weight decay 0.0001. We use a batch size of 96, and once again report the target test accuracy of the final model (after 100 epochs).

On BREEDS Entity-30, which contains 300K examples, we instead linear probe.

## B  Results and Protocol for Estimating $(\alpha, \beta, \gamma)$ – connectivity

Table 3 reports the connectivity estimates for the input space and the feature spaces learned by ERM, DANN, and SwAV for all pairs of DomainNet domains.

To estimate the connectivity between 2 class-domain pairs $(i, d)$ and $(j, d')$, we use the following algorithm:

1. Label all training examples of class and domain $(i, d)$ as 0 and all training examples of $(j, d')$ as 1. Discard the remaining examples.

2. Train a ResNet50 for 100 epochs using SwAV augmentations and cosine learning rate. We did not exhaustively tune this training step, but we kept the hyperparameters constant for all the pairs.

3. Collect the test set analogously to step 1, and evaluate the classifier on augmented data.

Each domain in DomainNet-S has a unique label distribution (all of which are far from uniform), and therefore in computing the average connectivity we compute the weighted mean, where each pair of (class, domain) pairs is weighted by the ratio of the less to more frequent label (0 or 1).

## C  Simulation Details

To obtain an even more fine-grained picture, we run simulations on toy data settings, where we can precisely vary each type of connectivity, and observe:

1. Sharp thresholding behavior when $\alpha$ and $\gamma$ cross over.

2. $\alpha$ must be nontrivial.

The simulation is on a finite population so that we can study the spectral decomposition directly; however, later we show empirically that these ideas still hold in real-world, continuous-population settings. For each domain-class pair $dc$, we generate $N$ points in augmentation space. These augmented points inherit the domain-class assignments of their corresponding natural points (we have to ensure that all

| Input space | Different class, same domain, source | Different class, same domain, target | Same class, different domain | Different class, different domain |
|---|---|---|---|---|
| Real ↔ Sketch | 24.49 | 38.12 | 21.47 | 20.51 |
| Real ↔ Painting | 24.49 | 33.71 | 33.62 | 28.07 |
| Real ↔ Clipart | 24.49 | 35.20 | 23.12 | 21.18 |
| Sketch ↔ Painting | 38.12 | 33.71 | 24.34 | 23.16 |
| Sketch ↔ Clipart | 38.12 | 35.20 | 30.23 | 27.72 |
| Painting ↔ Clipart | 33.71 | 35.20 31.36 | 30.09 | |
| Avg. | 30.57 | 35.19 | 27.36 | 25.12 |

| SwAV pre-trained space | Different class, same domain, source | Different class, same domain, target | Same class, different domain | Different class, different domain |
|---|---|---|---|---|
| Real ↔ Sketch | 3.08 | 6.92 | 4.73 | 1.74 |
| Real ↔ Painting | 3.07 | 7.43 | 11.79 | 2.23 |
| Real ↔ Clipart | 3.02 | 6.34 | 8.93 | 1.73 |
| Sketch ↔ Painting | 8.50 | 10.65 | 5.55 | 2.66 |
| Sketch ↔ Clipart | 8.54 | 8.15 | 8.10 | 3.29 |
| Painting ↔ Clipart | 10.22 | 8.40 | 6.13 | 3.12 |
| Avg. | 6.07 | 7.98 | 7.54 | 2.46 |

| ERM feature space | Different class, same domain, source | Different class, same domain, target | Same class, different domain | Different class, different domain |
|---|---|---|---|---|
| Real → Sketch | 8.40 | 13.32 | 9.99 | 6.43 |
| Real → Painting | 8.40 | 16.24 | 22.23 | 8.87 |
| Real → Clipart | 8.40 | 13.15 | 14.16 | 6.63 |
| Sketch → Real | 12.46 | 11.75 | 16.36 | 8.45 |
| Sketch → Painting | 12.46 | 18.57 | 16.51 | 9.38 |
| Sketch → Clipart | 12.46 | 14.45 | 22.30 | 10.90 |
| Painting → Real | 22.59 | 13.45 | 26.50 | 14.10 |
| Painting → Sketch | 22.59 | 20.43 | 14.03 | 11.28 |
| Painting → Clipart | 22.59 | 18.73 | 19.55 | 14.20 |
| Clipart → Real | 17.17 | 15.35 | 21.18 | 11.48 |
| Clipart → Sketch | 17.17 | 21.45 | 17.47 | 12.03 |
| Clipart → Painting | 17.17 | 25.30 | 19.80 | 14.52 |
| Avg. | 15.16 | 16.85 | 18.34 | 10.69 |

| DANN feature space | Different class, same domain, source | Different class, same domain, target | Same class, different domain | Different class, different domain |
|---|---|---|---|---|
| Real → Sketch | 6.09 | 9.98 | 9.87 | 4.53 |
| Real → Painting | 6.27 | 13.07 | 21.55 | 6.72 |
| Real → Clipart | 6.36 | 9.82 | 14.37 | 4.98 |
| Sketch → Real | 8.50 | 7.53 | 15.38 | 5.71 |
| Sketch → Painting | 8.75 | 13.24 | 16.61 | 6.91 |
| Sketch → Clipart | 8.80 | 10.43 | 21.15 | 7.76 |
| Painting → Real | 14.93 | 7.70 | 22.14 | 7.46 |
| Painting → Sketch | 16.80 | 14.57 | 15.49 | 9.76 |
| Painting → Clipart | 14.78 | 11.31 | 13.24 | 7.35 |
| Clipart → Real | 11.89 | 10.64 | 17.32 | 7.08 |
| Clipart → Sketch | 12.40 | 16.18 | 16.09 | 8.90 |
| Clipart → Painting | 11.51 | 19.06 | 15.90 | 10.44 |
| Avg. | 10.59 | 11.96 | 16.59 | 7.30 |

Table 3: Empirical estimates of the different parameters of connectivity in the input space (top) and feature spaces computed by SwAV (second), ERM (third), and DANN (bottom). The numbers provided are the separation error.

the augmented points are distinct). Then for any new natural point $\overline{x} \in \overline{\mathcal{X}}$ that we want to augment, we simply choose one of our existing augmented points $x$ at random and designate $x$ as the augmentation of $\overline{x}$. Since we have assigned domain-class memberships to this base set of augmentations, we can define the connectivity of two domain-class pairs $(d_1 c_1, d_2 c_2)$ to be the probability that a natural point from $d_1 c_1$ is "augmented into" an augmented point in $d_2 c_2$ (and vice-versa). This probability is something we can easily change in the simulations to vary this notion of connectivity.

## D    ADDITIONAL DETAILS FOR SECTION 4.4

As another method for evaluating the connectivity measures (in addition to Figure 4), we inspect the weights learned from a linear regression model from 3 features $[\alpha, \beta, \gamma]$ to the target accuracy (de-biased). The weights are $[-6.94, 5.61, 0.87]$, which gives a large negative weight to $\alpha$ (different-

class-different-domain connecitivity), large positive weight to $\gamma$ (same-class-different-domain), and a small positive weight to $\beta$ (different-class-same-domain) as expected from our intuitions.

The following table contains more detailed results on the cosine similarity between domain and class classifiers.

|  | Class (domain 1) vs. Class (domain 2) | Class (domain 1) vs. Domain | Class (domain 2) vs. Domain |
|---|---|---|---|
| **Living-17** | | | |
| Source ↔ Target | 0.3971 | 0.0125 | 0.0158 |
| **DomainNet** | | | |
| Real ↔ Sketch | 0.1999 | 0.0143 | 0.0165 |
| Real ↔ Painting | 0.2280 | 0.0157 | 0.0166 |
| Real ↔ Clipart | 0.2222 | 0.0148 | 0.0149 |
| Sketch ↔ Painting | 0.1758 | 0.0146 | 0.0200 |
| Sketch ↔ Clipart | 0.1590 | 0.0155 | 0.0178 |
| Painting ↔ Clipart | 0.1383 | 0.0178 | 0.0221 |

Table 4: Cosine similarity of class and domain classifiers trained on SwAV representations (average over all classes). Class classifiers trained on the source and target individually learn similar linear weights, evidenced by the cosine similarity in the range 0.13 to 0.33. However, domain classifiers learn linear weights that are nearly orthogonal to the class classifier weights, suggesting that SwAV pre-training learns features containing domain and class information in somewhat separate directions.

