# OpenReview forum: "How does Contrastive Pre-training Connect Disparate Domains?"
_ICLR.cc/2022/Conference — ICLR 2022 Submitted_

### Official Review · Reviewer_pgHb · 2021-10-30

**Correctness:** 3
**Technical Novelty And Significance:** 3
**Empirical Novelty And Significance:** 3
**Recommendation:** 5
**Confidence:** 4

**Main Review:**

## Strengths
+ The topic of the study is of great interest, and to me is pretty novel and not explored in the literature.
+ The observations are intriguing: contrastive learning based methods connect disparate domains in a way that is different from traditional adversarial DA perspective. The findings could have implications on when / how contrastive learning works when data are from different domains.
+ The evaluation and presentation is clear and thorough. The empirical results are well aligned with the proposed connectivity model, which could be promising for understanding the behavior of CL.
+ The writing is clear and easy to follow; the paper is well structured.

## Main Weaknesses
1. __Ambiguous shift assumptions throughout paper__
- Throughout the paper, the authors never explicitly define what is the __shift assumption__ across domains. Is the study conducted under the covariate shift assumption (i.e., $p_i(x) \neq p_j(x)$ and $p_i(y|x) = p_j(y|x)$)? If so, what will be the results under other assumptions (e.g., label shift)? Will this affect the main conclusion?

2. __Investigations limited to visual recognition tasks__
- First, the title and body texts seem to hold without specifying the data modality. However, all experiments are limited to image data for visual recognition tasks. Nevertheless, CL has been successful beyond image data, for example, texts [1], audios [2] and time series [3]. The authors only experimented on image data, which might not be enough for a rigorous conclusion.
- The limited experiments only on image data also lead to doubts on whether some claims are true or not. For example, the paper claims "Contrastive pre-training is a strong domain adaptation method". This might be true for image data, as it is interpretable by human, and people can design effective augmentation methods (e.g., those in SimCLR). However, does this hold also for other data modalities, especially those without rich priors on how to do augmentation in the first place?

3. __Minor: only single source & single target domain considered__
- One minor drawback is that the paper considered only single source & single target domain. For self-supervised pre-training, uncurated data is more likely to come from different domains [4]. The analysis under such case however is missing. It would be helpful to discuss the differences / similarities when there are multiple source domains exist.

4. __Minor: no actual algorithms / application scenarios provided__
- I understand that as a starting point, the paper only analyzed the behavior for CL across disparate domains. It would be nice to also give practical algorithms on how should one select (sub)sets of unlabeled data for better transfer. The analysis in the paper is good; however, I find it a bit disconnected with practical application scenarios.

## Other issues / questions
- Since the overall objective is agnostic to the data modality, is the proposed scheme also be extended to other data domains beyond images, like time series / texts? What would be the most challenging parts when extending to other modalities?
- How much is the performance dependent on the quality/effectiveness of the augmentation set $\mathcal{A}$?

## References
[1] SimCSE: Simple Contrastive Learning of Sentence Embeddings.

[2] Representation Learning with Contrastive Predictive Coding.

[3] Unsupervised Representation Learning for Time Series with Temporal Neighborhood Coding.

[4] Divide and Contrast: Self-supervised Learning from Uncurated Data.

**Summary Of The Paper:**

The paper studies how contrastive learning (CL) behaves in the unsupervised domain adaptation (UDA) setting. In particular, it finds that CL brings a different mechanism for UDA compared to traditional adversarial DA methods, where the features learned are far apart between the source and target domains. It further develops a conceptual model for explaining the success of contrastive learning under domain shifts, and empirically demonstrates through experiments on several benchmark datasets.

**Summary Of The Review:**

Overall, the idea is interesting and intriguing, however there are still limitations and ambiguities in the current draft. The detailed comments/questions are listed in the weaknesses / questions part.

The paper has a great potential to the CL field, and could be beneficial and inspiring for broader audience; but issues need to be addressed / made clear. I'm happy to change my score if the feedback addresses my concerns. Please refer to the points in the weaknesses / questions part. I would like to see feedbacks on these comments/questions.

---

> ### Author Response · Authors · 2021-11-19
> **Response to Reviewer pgHb**
>
> We thank reviewer pgHb for the detailed and insightful comments. We appreciate the reviewer’s feedback that the paper has “great potential to the CL field, and could be beneficial and inspiring for broader audiences.” We address each of the concerns below:
>
> > “Throughout the paper, the authors never explicitly define what is the shift assumption across domains.”
>
> This is an important point and we apologize that the original presentation of our shift assumption was confusing. Because contrastive pre-training operates in a different way from prior domain adaptation, our shift definition does not easily fit into an existing framework. Concretely, we assume that:
> 1. Every input has a single label and the source and target have identical label sets---we have clarified this in the revision.
> 2. The 3 parameters in definition 2 capture the relevant connectivity quantities that govern the success of contrastive learning.
> Because the assumption is on connectivity via data augmentation and not explicitly on the raw input space, we believe this definition can capture a wide variety of shifts, including covariate and label shift.
>
> Empirically, the DomainNet dataset has extreme label shifts across domains (the label distributions are illustrated in figure 11 of [1]), and therefore prior works (such as the SENTRY algorithm) require heuristics to handle the label shift. The strong empirical results of SwAV on DomainNet show that **contrastive pre-training handles label shift without any modification**, which we believe is a strength. In the setting of class imbalance, concurrent work [2] showed that self-supervised learning is more robust than ERM to class imbalance, but they do not study the domain adaptation setting.
>
> > “Since the overall objective is agnostic to the data modality, is the proposed scheme also be extended to other data domains beyond images, like time series / texts? What would be the most challenging parts when extending to other modalities?”
>
> For the experiments in this work we focus on pre-training for visual adaptation tasks. We have updated the paper to reflect that we consider only image data for the experiments. We thank the reviewer for bringing this up and we apologize for not making the data modality clear in the original submission.
>
> Good data augmentations are critical to many UDA methods, and therefore for a given modality the choice of the augmentation will likely influence both the joint-training and pre-training methods. With this in mind, contrastive learning leverages those augmentations differently from other UDA methods. For example, SENTRY uses augmentations for psuedolabel consistency regularization during entropy minimization, while contrastive learning uses the connectivity graph that the augmentations induce.
>
> > “Minor: only single source & single target domain considered”
>
> In table 1 we report results with pre-training on data from additional related domains, and we find that the additional data consistently improves target performance. For BREEDS, we use the entirety of ImageNet as the additional data, and for DomainNet we pre-train on all 4 domains.
>
> > “Minor: no actual algorithms / application scenarios provided”
>
> - We propose a simple method for domain adaptation: 1. Pretrain on source + target unlabeled data, 2. Fine-tune on the source labeled data. Without any tuning this is competitive with strong UDA methods. We believe this is novel, and the **simplicity and strong performance makes it a very practical method.**
> - Data selection: Given a very large pretraining dataset it is too expensive to pretrain on all examples and we need to select the most useful examples to pretrain on [3]. Our experiment in Section 4.4 suggests a method for selecting what examples to pretrain on. Specifically, we showed that examples that are "in between" two domains are the most important---removing them substantially hurts accuracy.
>
> In section 4.4 we **remove subsets of data based on our connectivity model that reliably reduces the target accuracy of pre-training on the Living-17 dataset.** In particular,identification of the “most connected” data points--i.e., the examples nearest the decision boundary of a domain classifier--significantly affects the target accuracy. This suggests a data selection algorithm based on connectivity for selecting useful data for contrastive pre-training.
>
> [1] SENTRY: Selective Entropy Optimization via Committee Consistency for Unsupervised Domain Adaptation. V Prabhu et al. ICCV 2021.
>
> [2] Self-supervised Learning is More Robust to Dataset Imbalance. Anonymous submission to ICLR 2022.
>
> [3] Language Models are Few-Shot Learners. TB Brown et al. NeurIPS 2021.

---

> > ### Comment · Reviewer_pgHb · 2021-11-23
> > **Thanks for the reply; several questions remained**
> >
> > I thank the authors for their detailed reply. I still have several remained questions regrading the rebuttal.
> >
> > __[Shift assumptions.]__
> >
> > I agree that contrastive pre-training operates in a different way as DA methods, however, this does not necessarily mean the assumptions do not fit --- the assumption is independent of the actual technique you apply, and which in your case might even be more important, as it characterizes when do the observations in the paper actually apply.
> >
> > If you assume only label shift exists here, then there is no concept shift across domains, and the setting degenerates to the class imbalance problem. However, for this setting, prior work [a] already showed _contrastive pre-training handles label shift without any modification_. This point is complemented by the experiments in this paper, but not a new finding.
> >
> > [a] Rethinking the value of labels for improving class-imbalanced learning. NeurIPS 2020.
> >
> > > The 3 parameters in definition 2 capture the relevant connectivity quantities that govern the success of contrastive learning. Because the assumption is on connectivity via data augmentation and not explicitly on the raw input space, we believe this definition can capture a wide variety of shifts, including covariate and label shift.
> >
> > I'm not sure what you mean by "assumption is on connectivity via data augmentation". The shifts across domains are decided by the data generation process, not the data augmentation you apply afterwards.
> >
> > __[Augmentation related.]__
> >
> > Contrastive learning requires effective augmentations to work. Based on the author response, I'm less convinced that the framework can be extended to other modalities where we don't have augmentation priors. Thus, it is important to highlight in title/abstract that the observations only apply to visual data.
> >
> > Even for visual data, different augmentations can lead to potentially substantial different results. Consider a toy case where the shift between domains is the pixel-wise color distribution sihft. If no "Color Jitterring" applied for CL, it will easily learn a shortcut solution (see experiments in the SimCLR paper). However, add this augmentation or not will not affect the performance of DANN-based methods which eliminate spurious correlation. This again highlight that data augmentations acts as a different role for UDA and CL, and is more crucial for CL when applied to data from multipe domains.
> >
> > A follow up question: How much is the performance dependent on the quality/effectiveness of the augmentation set $\mathcal{A}$?
> >
> > __[Single source & single target domain considered.]__
> >
> > If I understand correctly, although the experiments are performed over multiple domains, the connectivity model (the 3 parameters), as well as all the analysis (e.g., Figure 4) is based on only single source/target (i.e., only 2 domains considered). What if multiple domains considered here?

---

> > > ### Author Response · Authors · 2021-11-23
> > > **Modern theory of distribution shifts uses different notions of shifts**
> > >
> > > **[Shift assumptions.]**
> > >
> > > This is a great question.
> > >
> > > In short, **modern ML datasets are very high dimensional and the source and target have disjoint supports**. Classical notions like covariate shift often don't make sense in these settings. The key challenge now is what structures and shift assumptions we can leverage to adapt to distribution shifts in high dimensions.
> > >
> > > We can see the problem with "covariate shift" here with a toy example: covariate shift means that P(Y | X) is the same for the ID and OOD distributions. But **if P_id and P_ood have disjoint supports, then covariate shift can always be true** since any x in OOD never occurs in ID.
> > >
> > > This has **spurred a search for how we can adapt to distribution shifts in the high dimensional setting**. We need to make a different set of assumptions about the type of shift. Here's one example: https://arxiv.org/abs/2010.03622 (ICLR 2021 Oral) which assumes two things about the shift: 1. the source classifier has reasonable accuracy on the target, and 2. the target distribution satisfies the expansion assumption. Other works like (https://arxiv.org/pdf/2002.11361.pdf) assumes that the shift is gradual in a Wasserstein metric but don't explicitly assume notions of covariate shift or label shift, so that they can handle the disjoint support issue.
> > >
> > > In our case our assumption is in the style of the above self-training paper: the shift can be anything, so long as it satisfies the "connectivity" assumption under augmentations. We understand that this is different from the classical statistical notion of defining covariate shift, label shift, etc, but we believe this sort of assumption is necessary to deal with shifts in high-dimensional statistics, and pretty representative of the types of assumptions in the modern theory of distribution shifts (as opposed to older works like kernel mean matching, importance weighting, etc).
> > >
> > > **[Augmentation related.]**
> > >
> > > We agree with this point - we do need good priors on the augmentations. Thank you for bringing this up. We have edited the abstract and paper to emphasize vision more, and will add a discussion of this as well. A different choice of augmentations would change the connectivity parameters in definition 2, and will affect the performance through that - so our framework could give a way to decide what augmentation set to use.
> > >
> > > **[Single source & single target domain considered.]**
> > >
> > > We don't believe that multiple domains pose a fundamental challenge, however we thought it was cleanest to illustrate the insights with a single source and single target.

---

> > > > ### Comment · Reviewer_pgHb · 2021-11-26
> > > > **You need to be specific about what you mean by distribution shift**
> > > >
> > > > - Assumptions on distribution shift
> > > >
> > > > I agree with the authors that one might not classify the considered shifts into traditional assumptions. However, you still need to be very specific about what the shift is, and why it is important / can reflect the real cases, otherwise the results or claims you made are not interpretable.
> > > >
> > > > Further, from the experiments and descriptions in current paper, the actual support is the same across all domains, so there's nothing related to "source and target have disjoint supports" considered in this paper. From this perspective, I do believe you can categorize your scheme into one of the classic shift assumptions.
> > > >
> > > > - Augmentation related
> > > >
> > > > Thanks for your reply. Though it would be interesting to see the actual performance using different augmentation sets with varying strengths.
> > > >
> > > > - Considering multiple domains in analysis and connectivity model
> > > >
> > > > While I agree that using a single source and single target would be a good start, I'm not convinced that multiple domains are similar as the single one. When multiple training domains exist, their connectivities with the target domain (or even multiple target domains) would be different -- in other words, there would be a "distance" measure between source domains & target domain. In such cases, how to effectively model and utilize the "tightly" or "loosely" connected domains are crucial for successful transfer. For example, in DomainNet experiments, when using 3 domains as sources (and 1 as target), the 3 sources will have different connectivity measures w.r.t. the target one. How to model the connectivity model across such multiple-domain settings are more important and practical.
> > > >
> > > > ---------------------
> > > > In all, I do believe this paper has a great potential, but to me the paper in its current form is incomplete. I encourage the authors to incorporate the discussions and improve the paper.

---

> > > > > ### Author Response · Authors · 2021-11-26
> > > > > **Response**
> > > > >
> > > > > > Further, from the experiments and descriptions in current paper, the actual support is the same across all domains
> > > > >
> > > > > By "support", we mean the "set of points that have probability > 0". For sketch to real images, for example, even though they have the same input shape, they have disjoint support since sketches are grayscale and don't look like real images. In these cases, covariate shift is trivial to satisfy. For example, suppose we have two domains D1 and D2. In domain D1, there is only one possible point X1. Similar in domain D2, there is only one possible point X2 $\neq$ X1. Covariate shift is  trivial to satisfy, since covariate shift only constrains P(Y | X) to be the same across domains *for the same X*. Thus, *any label distribution P satisfies covariate shift* under disjoint supports, even while P(Y | X1) and P(Y | X2) may be *arbitrarily different*.
> > > > >
> > > > > >  you still need to be very specific about what the shift is, and why it is important / can reflect the real cases
> > > > >
> > > > > We agree and will clarify this. Since every input has a single label (which doesn't change across domains), we satisfy the covariate shift assumption. However, based on our discussion in the previous point, the covariate shift assumption is not informative about the relation between two domains in real cases where the supports are disjoint and doesn't constrain the space of shifts much. The other specific assumption we make is that the augmentation-based connectivity conditions must be satisfied between the domains. We use this as a way to better understand this setting where the domains are disparate.
> > > > >
> > > > > > How to model the connectivity model across such multiple-domain settings are more important and practical.
> > > > >
> > > > > We agree that modeling multiple domain settings is important. Since pretrained models usually pool together all the available data for pretraining, we can view multiple source domains as a single pooled source domain. From our current understanding, the connectivity measures between this pooled source domain and the target domain is what is important - certain source domains may contribute positively or negatively to the pooled connectivity measures.

---

### Official Review · Reviewer_W3eA · 2021-11-02

**Correctness:** 3
**Technical Novelty And Significance:** 2
**Empirical Novelty And Significance:** 2
**Recommendation:** 3
**Confidence:** 3

**Main Review:**

Strong points of the paper:
1.	The goal of applying contrastive pre-training on unsupervised domain adaptation is a critical topic to study.
2.	The paper proposes to use the connectivity model to study contrastive learning.
3.	Experiments are comprehensive with a variety of benchmark tasks.

Weak points of the paper:
1.	This paper does not propose a new method, but proposes a connectivity measure based on the analysis of existing methods. Despite it maybe practical to improve fine-tuning with connectivity measure, the technical contribution is rather limited.
2.	The main experiments in this article are based on existing methods. It would be good to show the results of using domain adaptation methods based on the new connectivity measure. Even a simple solution on the toy task can verify the practicality of the proposed measure to a certain extent.


**Summary Of The Paper:**

This paper proposes to study the contrastive pre-training on domain adaptation tasks. The key contribution is to design a connectivity model for better data selection and data augmentations of pre-training. To verify the benefit of contrastive pre-training to domain adaptation, experiments are performed on a variety of benchmark tasks including BREEDS, DomainNet and CIFAR-10.

**Summary Of The Review:**

Although this paper gives an interesting new idea to solve the problem of domain adaptation from connectivity, it does not extend the idea to a specific method, nor does it conduct sufficient experimental verification.

---

> ### Author Response · Authors · 2021-11-19
> **Response to Reviewer W3eA**
>
> We thank reviewer W3eA for the comments and we appreciate their feedback that our paper presents “an interesting new idea to solve the problem of domain adaptation from connectivity.” We address their concerns below:
>
> > "paper does not propose a new method"
>
> - We propose a simple method for domain adaptation: 1. Pretrain on source + target unlabeled data, 2. Fine-tune on the source labeled data. Without any tuning this is competitive with strong UDA methods. We believe this is novel, and the simplicity and strong performance makes it a very practical method.
> - Our paper focuses on **understanding the mechanism** by which contrastive pre-training adapts to new domains, and therefore our analysis with the connectivity model is focused on relating the connectivity structure of the data to the downstream target accuracy. Furthermore, one purpose of our toy task is to illustrate that **out-of-the-box pre-training generalizes well to the target precisely when our connectivity assumption is satisfied.**
> - Data selection: Given a very large pretraining dataset it is too expensive to pretrain on all examples and we need to select the most useful examples to pretrain on [1]. Our experiment in Section 4.4 suggests a method for selecting what examples to pretrain on. Specifically, we showed that examples that are "in between" two domains are the most important---removing them substantially hurts accuracy.
>
> > “Even a simple solution on the toy task can verify the practicality of the proposed measure to a certain extent.”
>
> In our paper, we showed that contrastive learning solves our toy task perfectly whenever the connectivity assumption is satisfied, via the following mechanism:
> 1. The pre-training step learns a feature space in which the class and domain information are disentangled into separate dimensions.
> 2. A linear classifier trained on top of the pre-trained features can perfectly classify the source examples using only the dimensions containing class information, thereby generalizing well to the target.
>
> Our connectivity measure suggests a data selection algorithm for selecting new pretraining data. Suppose the toy task did not satisfy the connectivity conditions -- we can use connectivity to select data to add to the pretraining dataset to improve target accuracy. This is essentially what our data selection experiment in Section 4.4 is testing.
>
> [1] Language Models are Few-Shot Learners. TB Brown et al. NeurIPS 2021.

---

### Official Review · Reviewer_6UNN · 2021-11-02

**Correctness:** 4
**Technical Novelty And Significance:** 3
**Empirical Novelty And Significance:** 3
**Recommendation:** 5
**Confidence:** 4

**Main Review:**

- The paper is mainly about a finding that pretraining with SwAV ( a recently proposed self-supervised method) can benefit unsupervised domain adaptation tasks. This is done by pretraining on both source and target models.
- Overall, the findings are interesting. But it would be more convincing if the authors report the results of initializing the backbone used in standard UDA tasks with pretrained self-supervised model (SwAV) in this case. So this would make comparisons with other approaches more fair.
-  It would be more convincing if the authors experiment with other self-supervised models like MOCO, SIMCLR. I am curious whether the same conclusion would hold.
- The presentation could be further improved---there are lots of typos...

**Summary Of The Paper:**

The paper investigates why contrastive learning method benefits unsupervied domain adaptation.  In particular, the authors use SwAV, a recently proposed self-supervised learning method to pretrain a model on both targeted and source datasets. The authors found that by using such a model for UDA tasks surprisingly nice results are achieved. The authors analyze the reasons behind using a connectivity model.

**Summary Of The Review:**

Overall, I think the observations in this paper are interesting. But I think more experiments would make the paper stronger.

---

> ### Author Response · Authors · 2021-11-19
> **Response to Reviewer 6UNN**
>
> We thank reviewer 6UNN for the review and we appreciate that they find the observations in our paper interesting. We address the concerns below:
>
> > “It would be more convincing if the authors report the results of initializing the backbone used in standard UDA tasks with pretrained self-supervised model (SwAV) in this case.”
>
> This is an interesting idea; it is likely that running a standard UDA method on top of the SwAV pre-trained features would improve over the simple fine-tuning method that we currently use. However, training both algorithms sequentially would double the computation requirements, and thus for a fair comparison we compare joint-training and pre-training where both are initialized randomly. Furthermore, the goal of our experiments is to quantify the inherent robustness in contrastive learning methods without the aid of explicit domain adaptation mechanisms.
>
> For more in-depth comparison with standard UDA methods, we have added experiments with domain adversarial neural networks (DANN) [1] as another baseline on BREEDS and DomainNet. We find that SwAV matches or outperforms both SENTRY and DANN on all our datasets, and the results are reported in table 1 (copied in the general response above).
>
> > “It would be more convincing if the authors experiment with other self-supervised models like MOCO, SIMCLR.”
>
> For diversity in the contrastive methods, we provide the results of SimCLR on the STL $\to$ CIFAR task in table 1, showing that it outperforms the Dirt-T baseline---so our results are not specific to SWaV but also apply to SimCLR.
>
> Finally, we apologize for the typos in the original submission; we have now fixed them and clarified the presentation throughout the paper.
>
> [1] Domain-Adversarial Training of Neural Networks. Y Ganin et al. JMLR 2016.
>
> [2] A Theory of Label Propagation for Subpopulation Shift. T Cai et al. ICML 2021.

---

### Official Review · Reviewer_tPmk · 2021-11-02

**Correctness:** 4
**Technical Novelty And Significance:** 2
**Empirical Novelty And Significance:** 2
**Recommendation:** 5
**Confidence:** 3

**Main Review:**

The main strength and contribution of the paper is the analysis of the source-domain connectivity, which was interesting to read. It was well illustrated through a toy example, and then applied to some real-life domain adaptation datasets. The whole analysis seems sound and well-though.
 However the main two weakness are **(i)** there is a lack of clarity on the data augmentation used for contrastive learning; these seem to be key to define the whole notion of connectivity so it is a bit surprising they're not discussed in more details; And **(ii)** it's not clear how significant the reported work is and what it could lead to; In particular, one of the claims of the paper is that contrastive pretraining can reach similar accuracies as other UDA techniques while not aiming to align the domains as adversarial UDA techniques do. However there is not much comparison to these adversarial techniques in the second part of the paper; For instance, Figure 5 suggest that, while domain embeddings can be easily separated with contrastive pretraining, there is still some kind of "domain connectivity" which is really important for final accuracy, which seems to actually match the "domain alignment" goal from adversarial UDA techniques.


**Some more precisions on parts of the paper I found a bit unclear:**
  * (i) **Unclear definition of the positive and negative sets for contrastive learning.** It is briefly mentioned in Section 3.2 that the positive sets in contrastive learned are defined from multi-crop data augmentations, however it's not really well detailed, and in particular it's not clear how this impacts final accuracy (e.g. how much does the features quality degrade with simpler data augmentations ? how much tuning does it require ?)
  * (ii) **Choices of datasets and baselines.** The experiments of Table 1 only seem to have one "fair" baseline (DirT/Sentry) because ERM is trained on the source data online. It would have been interesting to experiment on datasets such as Visda2017 or Office-31 that seem to be more commonly used in general and might have more benchmark results available.
  * (iii) **Impact of data augmentation**: This links to points (i) and (ii): It's not clear to me how much the data augmentation strategy impacts the model, and whether it could also improve the baselines: Are Sentry and ERM baselines also trained with the same data augmentation as the contrastive learning techniques ?
* (iv) **Readability of table 2** Table 2 and Section 4.1 would be stronger with more details on the classifiers used. For instance, if I understood correctly, the authors train a classifier to separate the source and target domain on either (i) the input images or (ii) the pretrained contrastive features. Because classifier (ii) as higher accuracy than (i), they conclude that the contrastive features

**Small typos/unclear notations:**
  * **Table 1, middle, second row**: the highest accuracy (Sentry) is not bold
  * **Definition 1** Shouldn't the definition of connectivity include some kind of expectation over drawn samples $x$ and $x'$ ?

**Summary Of The Paper:**

The authors tackle the problem of unsupervised domain adaptation (UDA), where most state-of-the-art methods aim to learn a joint embedding space for samples from the (labeled) source domain and (unlabeled) samples from the target domain.

In contrast to this, the authors propose to use contrastive learning pre-training techniques to first learn an encoder using target/source/additional data: Intuitively, this objective pushes positive samples together, and further away from other samples. The notion of "positive set" is defined as all samples which are obtained from data augmentations of a same sample natural image.
A classifier is then learned on top of these features to solve the classification task on the labeled source dataset.

Using standard contrastive learning techniques, the authors show that they can achieve similar accuracies as some SoTA baselines on three UDA datasets. The remaining of the paper analyses the features learned by this techniques, and in particular, how they do not learn to align the two domains as is the common intuition in UDA. This analysis relies on a "connectivity graph" defined for contrastive learning techniques, which, in the current setting, can be used to connect samples from the source and target domain, to determine how likely they are to be in the same positive set.

**Summary Of The Review:**

The main strength and contribution of the paper is the analysis of the source-domain connectivity, which was interesting to read. It was well illustrated through a toy example, and then applied to some real-life domain adaptation datasets. The whole analysis seems sound and well-though.
 However the main two weakness are **(i)** there is a lack of clarity on the data augmentation used for contrastive learning; these seem to be key to define the whole notion of connectivity so it is a bit surprising they're not discussed in more details; And **(ii)** it's not clear how significant the reported work is and what it could lead to; In particular, one of the claims of the paper is that contrastive pretraining can reach similar accuracies as other UDA techniques while not aiming to align the domains as adversarial UDA techniques do. However there is not much comparison to these adversarial techniques in the second part of the paper; For instance, Figure 5 suggest that, while domain embeddings can be easily separated with contrastive pretraining, there is still some kind of "domain connectivity" which is really important for final accuracy, which seems to actually match the "domain alignment" goal from adversarial UDA techniques.

---

> ### Author Response · Authors · 2021-11-19
> **Response to Reviewer tPmk**
>
> We thank reviewer tPmk for the detailed review. We appreciate that tPmk found the source-target connectivity analysis “interesting to read” and “well-illustrated through the toy example”. We address the concerns below:
>
> > “there is a lack of clarity on the data augmentation used for contrastive learning”
>
> We apologize for not being clear--we use the standard data augmentation strategy used by SwAV, which involves cropping, flipping, color jitters, etc. This is described in full detail in appendix A.2 of [1].  Note that **we use the contrastive pre-training methods (along with its default augmentations) out-of-the-box and do not tune the augmentations** for any dataset. We’ll make this clear in the paper.
>
> > “It's not clear to me how much the data augmentation strategy impacts the model, and whether it could also improve the baselines”
>
> This is a great point, and we have now run ERM and Domain adversarial neural networks (DANN) baselines on DomainNet both with standard augmentations and with the data augmentations used in SwAV. The results are in Table 1 (copied in the general response as well). **SwAV (44.91% average target accuracy) outperforms DANN with both standard augmentations (39.92%) and SwAV augmentations (43.99%).** The SENTRY joint-training baseline uses RandAugment, a strong augmentation method -- since SENTRY was developed and tuned using the DomainNet dataset and achieves SOTA results, we keep their augmentation pipeline as-is.
>
> > "not much comparison to these adversarial techniques"
>
> In the rebuttal period, **we have added DANN to our comparison and found that SwAV outperforms DANN both with standard augmentations and when both use the same augmentations** (numbers in the previous paragraph). To complement our estimates of domain distance in the pre-trained feature space, during the rebuttal period we also estimated the domain distance in the feature spaces learned by ERM and DANN. We found that **both baselines bring the domains close together, which is in stark contrast to SwAV** which pushes them far apart. These results are reported in table 2 (copied in the general response above).
>
> > Contrastive learning still depends on “some kind of 'domain connectivity' which is really important for final accuracy, which seems to actually match the "domain alignment" goal from adversarial UDA techniques.”
>
> We find empirically that contrastive pre-training learns features where the source and target domains are very far apart but also generalizes well between source and target. In contrast, domain adversarial methods align the domains, meaning that the learned features overlap. The underlying mechanism must be different since we find that contrastive learning does not bring the domains closer together.
>
> While on the surface “domain connectivity” and “domain alignment” seem similar, **contrastive learning works for a very different reason from domain adversarial methods**. Domain connectivity relies on the *relative* differences between different connectivity measures ($\alpha$, $\beta$, $\gamma$), while the *absolute* connectivities can be small. This allows for the features to be far apart, and we show that learning on source can still generalize well to target. Domain alignment is much stronger since it requires the domains to be indistinguishable.
>
> [1] Unsupervised Learning of Visual Features by Contrasting Cluster Assignments. M Caron et al. NeurIPS 2020.

---

### Author Response · Authors · 2021-11-19
**Copied Table 1 and 2 for easy access (part 1/2)**

We thank all 4 of our reviewers for their helpful and detailed feedback. In our review we ran additional experiments requested by the reviewers (an extra domain adaptation baseline DANN, and using SwAV augmentations with ERM and DANN). We find that SWaV still achieves comparable performance.

For easy reference, we have copied a summary of the test accuracies reported in Table 1, which we have also updated in the submission.

| ERM   | ERM (SA) | SENTRY | DANN  | DANN (SA) | SwAV  | SwAV+ |
|-------|----------|--------|-------|-----------|-------|-------|
| 24.20 | 32.22    | 32.81  | 39.92 | 43.99     | **44.91** | 51.73 |
---
|           | ERM   | SENTRY | DANN  | SwAV (S) | SwAV (T) | SwAV (S+T) | SwAV+ |
|-----------|-------|--------|-------|----------|----------|------------|-------|
| Living-17 | 59.00 | 68.76  | 66.88 | 61.89    | 68.53    | **73.59**      | 81.82 |
| Entity-30 | 50.97 | **62.80**  | 54.62 | 52.33    | 60.33    | 62.03      | 65.90 |
---
| ERM  | Dirt-T | SimCLR (T) |
|------|--------|------------|
| 63.6 | 75.3   | **76.1**       |

**Table 1:** (Top) Test accuracy (%) of ERM with standard and SwAV augmentations, SENTRY, DANN
with standard and SwAV augmentations, SwAV, and SwAV + extra (abbreviated to SwAV+) **on average** over all
domain pairs of DomainNet. Here, SwAV is run on source + target, and SwAV+ is not bolded because it uses additional data and is therefore not directly comparable. (SA) denotes SwAV augmentations. (Middle)
Test accuracy (%) of ERM, SENTRY, DANN, SwAV (source-only, target-only, and source + target splits), and SwAV+ on BREEDS.
(Bottom) Test accuracy (%) of ERM, Dirt-T, and SimCLR on STL-10 $\to$ CIFAR-10. ERM and Dirt-T
numbers are as reported from Shu et al. (2018).

---

> ### Author Response · Authors · 2021-11-19
> **Copied Table 1 and 2 for easy access (part 2/2)**
>
> While SwAV attains comparable performance with UDA methods, one of the key points of our paper is that SwAV works for a very different reason. Unlike methods like DANN which aim to bring the source and target domains close, SwAV actually pushes them further apart.
>
> To examine the claim that SwAV pushes the domains far apart, we computed the domain and class discrepancy for different methods. Abbreviated results on the DomainNet dataset are reported below and in full in Tables 2 and 3 of the paper.
>
> | | Different class,    | Different class,    | Same class,      | Different class, |
> |---------------|---------------------|---------------------|------------------|------------------|
> |    | same domain, source | same domain, target | different domain | different domain |
> | Input space   | 32.88               | 32.88                  | 27.36            | 25.12            |
> | DANN          | 10.59               | 11.96               | 16.59            | 7.30             |
> | SwAV          | 7.03                | 7.03                  | 7.54             | 2.46             |
>
> **Table 2:** Average separation error of DomainNet class-domain pairs. Classifiers were trained on augmented images to distinguish class-domain pairs as a proxy for connectivity (higher separation errors suggest greater connectivity).
>
> In particular, we find that
> 1. For SwAV: On both datasets, the domains are very distinguishable in the feature space (separable by a ResNet50 with only 7.54% error on DomainNet) learned by contrastive learning, and on DomainNet, the domains additionally remain as far apart as the classes (separable up to 7.03% error).
> 2. The DANN baseline differs from SwAV in two respects: (1) the domains are brought much closer together than the classes, and (2) the domains are less distinguishable in DANN, which makes sense since DANN optimizes for keeping the source and target domains close.

---

### Author Response · Authors · 2021-11-23
**General Response**

We thank all the reviewers for their detailed reviews. They found that our analysis of the source-target connectivity was “well-illustrated through the toy example” and that the “observations in this paper are interesting.” Some common issues raised, along with what we done to address them during the rebuttal, include:
- **Reviewers tPmk and pgHb pointed out that it is unclear how the chosen augmentation set influences the success of contrastive pre-training under domain shift.** In particular, the reviewers were concerned that (i) using different data augmentation for SwAV and the baselines might lead to unfair comparison and (ii) the reported conclusions may not hold when extending this work to modalities without standardized and high-quality augmentations.
    - **In the rebuttal, we have reported additional experiments with baselines that use the same augmentation as SwAV and find that SwAV is on par with or better than all baselines.** We have additionally updated the text to clarify that we focus on the visual domain adaptation setting.
- **Reviewers tPmk and 6UNN requested more thorough comparison with existing UDA methods, both in terms of empirical performance and in terms of the connectivity analysis.**
    - **In the rebuttal, we have added the DANN baseline to BREEDS and DomainNet and found that SwAV outperforms DANN.** To complement our estimates of domain distance in the pre-trained feature space, we additionally estimated the domain distance in the feature spaces learned by ERM and DANN and found that both baselines bring the domains close together. The results are reported in table 2 (copied in the comment below).
- **Reviewers W3eA and pgHb pointed out that there are no actual algorithms provided.**
    - In this work we propose a simple method for domain adaptation: 1. Pretrain on source + target unlabeled data, 2. Fine-tune on the source labeled data. Without any tuning we have found that this is competitive with strong UDA methods. We believe this is novel, and the simplicity and strong performance makes it a very practical method.
    - Our paper focuses on **understanding the mechanism** by which contrastive pre-training adapts to new domains, and therefore our analysis with the connectivity model is focused on relating the connectivity structure of the data to the downstream target accuracy. One important purpose of our toy task is to illustrate that **out-of-the-box pre-training generalizes well to the target precisely when our connectivity assumption is satisfied.**
    - Finally, our experiment in Section 4.4 suggests a method for selecting what examples to pretrain on. Specifically, we showed that examples that are "in between" two domains are the most important---removing them substantially hurts accuracy.

---

### Decision · Program_Chairs · 2022-01-20

**Decision:**

Reject

**Comment:**

This is a borderline paper. While reviewers believe the findings from this paper may be of potential interest, they are fully convinced. For instance, if the authors want to claim the proposed mechanism is general for UDA, then they should demonstrate its effectiveness to other application domain(s), such as the NLP domain, where the pretrain-finetuning strategy is widely adopted for transfer learning. However, the authors did not provide correspondingly additional experiments as requested by a reviewer but claimed they only focused on the CV domain. If the focus is on the CV domain, then the authors need to explain in detail why in the CV domain, the proposed mechanism works well (while in other domains, it may not). There are many other concerns about the assumptions, experimental settings, etc.

In summary, this is a borderline paper below the acceptance bar of ICLR.